# CBCT-to-CT Synthesis for Cervical Cancer Adaptive Radiotherapy via U-Net-Based Model Hierarchically Trained with Hybrid Dataset

**DOI:** 10.3390/cancers15225479

**Published:** 2023-11-20

**Authors:** Xi Liu, Ruijie Yang, Tianyu Xiong, Xueying Yang, Wen Li, Liming Song, Jiarui Zhu, Mingqing Wang, Jing Cai, Lisheng Geng

**Affiliations:** 1School of Physics, Beihang University, Beijing 102206, China; lx1634@buaa.edu.cn (X.L.); yang_xy@buaa.edu.cn (X.Y.); 2Department of Radiation Oncology, Cancer Center, Peking University Third Hospital, Beijing 100191, China; ruijyang@yahoo.com (R.Y.);; 3Department of Health Technology and Informatics, The Hong Kong Polytechnic University, Hong Kong SAR 999077, China; tian-yu.xiong@connect.polyu.hk (T.X.);; 4Beijing Key Laboratory of Advanced Nuclear Materials and Physics, Beihang University, Beijing 102206, China; 5Peng Huanwu Collaborative Center for Research and Education, Beihang University, Beijing 100191, China

**Keywords:** image enhancement, hierarchical training, artifacts removal, synthetic CT, adaptive radiotherapy, cervical cancer

## Abstract

**Simple Summary:**

Adaptive radiotherapy ensures precise radiation dose deposition to the target volume while minimizing radiation-induced toxicities. However, due to poor image quality and inaccurate HU values, it is currently challenging to realize adaptive radiotherapy with cone beam computed tomography (CBCT) images. Previous studies on CBCT image enhancement have rarely focused on cervical cancer and are often limited to training datasets from a single linear accelerator. The aim of this study is to develop a deep learning framework based on a hybrid dataset to enhance the quality of CBCT images and achieve accurate HU values, for application to cervical cancer adaptive radiotherapy. The synthetic pseudo-CT images generated by our model have accurate HU values and well-preserved edges of soft tissues. It can be further explored in multi-center datasets to promote its clinical applications.

**Abstract:**

Purpose: To develop a deep learning framework based on a hybrid dataset to enhance the quality of CBCT images and obtain accurate HU values. Materials and Methods: A total of 228 cervical cancer patients treated in different LINACs were enrolled. We developed an encoder–decoder architecture with residual learning and skip connections. The model was hierarchically trained and validated on 5279 paired CBCT/planning CT images and tested on 1302 paired images. The mean absolute error (MAE), peak signal to noise ratio (PSNR), and structural similarity index (SSIM) were utilized to access the quality of the synthetic CT images generated by our model. Results: The MAE between synthetic CT images generated by our model and planning CT was 10.93 HU, compared to 50.02 HU for the CBCT images. The PSNR increased from 27.79 dB to 33.91 dB, and the SSIM increased from 0.76 to 0.90. Compared with synthetic CT images generated by the convolution neural networks with residual blocks, our model had superior performance both in qualitative and quantitative aspects. Conclusions: Our model could synthesize CT images with enhanced image quality and accurate HU values. The synthetic CT images preserved the edges of tissues well, which is important for downstream tasks in adaptive radiotherapy.

## 1. Introduction

Cervical cancer ranks fourth in terms of commonly diagnosed cancers among women, and it is also the fourth leading cause of cancer-related deaths among women [1]. Despite the implementation of systematic screening programs to eliminate cervical cancer as a public health issue, the incidence and mortality rates of cervical cancer are still increasing for women who live in low-income countries with limited medical resources. Overall, both high-Human Development Index (HDI) and low-HDI countries still have high age-standardized rates in terms of cervical cancer. Concurrent chemoradiotherapy is the standard treatment for locally advanced cervical cancer [2]. In the course of treatment, it is crucial to ensure that sufficient doses are deposited in the planning target volume (PTV). On the other hand, the fewer organs at risk (OARs) that are irradiated the better. With the radiotherapy technique witnessing flourishing developments, image-guided systems are employed to ensure that the patient’s current treatment set-up follows the planning computed tomography (pCT), the so-called image-guided radiotherapy (IGRT) [3,4,5,6]. Before dose delivery, cone beam computed tomography (CBCT) is performed to provide submillimeter-resolution X-ray imaging of patient anatomy. By comparing the CBCT with pCT and adjusting patient set-up, one ensures that the patient is positioned correctly and that the radiation is being delivered to the intended target when the bony anatomy is well registered. Currently, only one individual treatment plan was made in advance and employed throughout the entire course for patient in a clinical routine. However, due to bladder filling, cervix motion, tumor regression, and gastrointestinal peristalsis [2,7,8], it is challenging to realize highly conformal dose delivery relying solely on the registration of bone structures. It is promising to realize adaptive radiotherapy by utilizing fractional CBCT to modify the treatment plan. Nevertheless, CBCT is generally less effective at visualizing soft tissues than pCT owing to different imaging principles. More photon scattering events occur within the patient’s body due to the use of the cone-shaped X-ray beam, resulting in reduced image contrast, large artifacts, and increased image noise in CBCT imaging [9]. Subsequently, the HU values in CBCT images often fluctuate and deviate significantly from the values obtained from CT scans [10]. Meanwhile, CBCT imaging uses a lower radiation dose than CT imaging, also resulting in low-quality images. In addition, the field of view is limited due to the technical constraints related to the design of the imaging system. These issues limit the clinical application of CBCT in adaptive radiotherapy.

Several methods have been proposed to correct the artifacts and improve the quality of CBCT images. These methods could be divided into hardware-based pre-processing methods [11,12,13,14] and image post-processing methods. Hardware-based methods attempt to decrease the influence caused by scattering by utilizing an anti-scatter grid when acquiring CBCT images [15,16]. Image post-processing methods mainly consist of deformation of pCT [17,18,19,20,21], an estimation of scatter kernels [22], Monte Carlo simulations of scatter distribution [23,24], histogram matching [25], and deep learning-based methods [26,27,28,29,30,31]. Deformation of pCT is one of the commonly used methods, which is based on deformable registration between pCT and CBCT. However, it is worth mentioning that there is no optimal deformable registration algorithm, which means that there will be extra uncertainties after registration, which subsequently affects the correctness of HU values in CBCT images. Estimations of scatter kernels and Monte Carlo simulations are time-consuming and resource-consuming. Histogram matching requires a special linear matching method and also relies on deformable registration.

Deep learning-based methods have developed rapidly and obtained promising results in recent years. Previous studies can be classified into scatter correction based on projection and synthetic CT generation [26]. The first method attempted to correct the scattering in the raw projection domain and then to enhance the image quality of CBCT. Nomura et al. [32], Rusanov et al. [33], Lalonde et al. [34], Hansen et al. [35], and Landry et al. [36] all used U-Net-based models to estimate the scatter signal in raw projections and then correct the intensity for CBCT images. The second method attempted to directly generate CT images with enhanced image quality from original CBCT images and was investigated in several anatomical sites. Qiu et al. [37] integrated histogram matching into a cycle generative adversarial network (Cycle-GAN) to generate synthetic CT images. The effectiveness of their model was demonstrated in the thoracic region. Yang et al. [38] developed a transformer-based model to enhance the image quality of CBCT in the head and neck. Tien et al. [39] combined Cycle-GAN with Deblur-GAN to generate more realistic images from CBCT. They compared their model with U-Net and achieved better HU value agreement in the thoracic region. Lemus et al. [40] assessed the performance of using pseudo-CT generated via Cycle-GAN to calculate dosimetry in the abdominal region. They reported that synthetic CT outperformed the deformable pCT both in anatomical structure similarity and dosimetry accuracy. Xue et al. [41] compared the performance of Cycle-GAN with GAN and U-Net in pseudo-CT generation for nasopharyngeal carcinoma radiotherapy and reported that Cycle-GAN outperformed the other two models in terms of HU mean absolute error (MAE) and anatomy-preserving capacity. Kida et al. [42] used U-Net with MAE as the loss function to generate pseudo-CT from CBCT. And then they also employed Cycle-GAN to translate CBCT into pseudo-CT and demonstrated the effectiveness of their model in prostate cancer patients [43]. Sun et al. [44] investigated the performance of 3D Cycle-GAN in a pseudo-CT generation task for cervical cancer patients. Wu et al. [45] utilized deep convolution neural networks with residual blocks (RCNN) to synthesize CT from CBCT in prostate cancer patients and obtained substantial improvement in the accuracy of HU values. We noted that Cycle-GAN-based models were frequently used for pseudo-CT synthesis tasks. While Cycle-GAN allows for training models with unpaired image data, it would sometimes generate structures in pseudo-CT images that actually do not exist in the original CBCT images. Moreover, the training speed is lower than other models and it requires more computation resources.

Previous studies related to the pelvic region mainly focused on prostate cancer instead of cervical cancer. The possible reason is that the organ motion in the cervical region is more complex compared to that in the prostatic region. Juneja et al. [46] reported that the mean motion during the treatment for prostate was 1.1 ± 0.9 mm. However, previous studies have shown that movements of the cervix and uterus can be up to 4–6 cm [47,48,49]. Furthermore, cervical cancer patients may have intra-uterine devices in their pelvic region. Intra-uterine devices may shift with the movement of the uterus. Therefore, it is more challenging to obtain well-paired image data to train the model for cervical cancer patients. Moreover, the models developed previously were trained with datasets acquired from a single linear accelerator. Different linear accelerators may have different acquisition protocols, especially if they are made by different vendors, resulting in variations in raw image quality. Furthermore, each linear accelerator may have its own unique noise distribution due to variations in hardware components. Thus, it is more challenging to develop a robust model based on a hybrid dataset.

In this study, we attempted to complete the pseudo-CT synthesis task using models trained with paired image data selected from the pelvic region for cervical cancer treatment. We adopted U-Net as the baseline model and integrated residual blocks into each convolution layer. A hierarchical training strategy was utilized to train the model progressively from a coarse resolution to a fine resolution. The effectiveness of this strategy has been proven in image processing tasks [45,50]. Our purpose is to alleviate the artifacts, enhance the visuality of soft tissues, and acquire more accurate HU values in CBCT images. We attempted to train a robust model with a hybrid dataset obtained from different brands of linear accelerators and compared it with models trained with a single dataset. To demonstrate the effectiveness of our deep learning-based framework, we compared its performance with an RCNN and a residual U-net not trained with a hierarchical training strategy.

Our study mainly makes contributions in the following aspects:A specially tailored deep learning framework that combined residual U-Net with a hierarchical training strategy was constructed and optimized to synthesize high-quality pseudo-CT images with accurate HU values and enhanced clarity from original CBCT images.A combination of weighted MAE, multi-scale structural similarity index (MS-SSIM) loss, and gradient difference loss was utilized to constrain the training process to effectively preserve the edges of tissues.A hybrid dataset collected from different brands of linear accelerators was used to construct a more robust model and demonstrate its generalizability.

## 2. Materials and Methods

### 2.1. Data Collection and Image Pre-Processing

#### 2.1.1. Data Collection

In our study, CBCT images and corresponding pCT images of 230 cervical cancer patients were collected. All the patients were diagnosed with primary cervical cancer and received concurrent chemoradiotherapy during 2012–2021. All the patients were Chinese females. The age range of these patients is between 32 and 89, and the mean age is 58 years old (standard deviation: 12). All the pCT images were acquired with a Brilliance Big Bore CT scanner (Philips Healthcare, Best, The Netherlands), with a tube voltage of 120 kV, an exposure of 300/325 mAs, an image resolution of 512 × 512, a pixel size of 0.98 × 0.98 mm^2^, and a slice thickness of 5 mm. Fifty patients were treated with Elekta Axesse LINAC (Elekta, Stockholm, Sweden) and the CBCT images were acquired using the XVI on-board CBCT imaging system with a tube voltage of 120 kV, a tube current of 80 mA, an image resolution of 410 × 410, a pixel size of 1 × 1 mm^2^, and a thickness of 1 mm. The other 180 patients were treated with Varian Trilogy LINAC (Varian, Palo Alto, CA, USA) and the CBCT images were obtained using the OBI imaging system. The scanning parameters were as follows: a voltage of 125 kV, a tube current of 80 mA, an image resolution of 512 × 512, a pixel size of 0.88 × 0.88 mm^2^, and a thickness of 2.5 mm.

We manually checked the collected dataset again and finally 228 patients were enrolled in this study (49 patients obtained from Elekta Axesse and 179 patients obtained from Varian Trilogy). More details related to our criteria are shown in Figure 1.

#### 2.1.2. Image Pre-Processing

Given that the pCT images and CBCT images were scanned at different times, pCT images were first registered to the CBCT images using MIM Software (v.6.3.4, MIM Software Inc., Cleveland, OH, USA). Then, in order to mitigate the interference of non-anatomical structures such as the couch, cotton swabs, and intra-uterine devices, several steps were adopted to remove them from images and other anatomical structures were preserved. Firstly, body masks were generated by using the tool TotalSegmentator developed by Wasserthal et al. [51]. TotalSegmentator was based on nnUNet [52], trained on diverse CT images, and performed well in most scenarios. We manually checked the body mask after running TotalSegmentator and used the operation of erosion and dilation to confirm that the body contour was smooth. Next, pCT and CBCT images were multiplied by the body mask to remove the couch. Cotton swabs with a contrast agent were inserted into the vagina to help the radiation oncologist locate the cervix more clearly. However, the cotton swab was removed when delivering the dose, resulting in mismatch between pCT and CBCT in some regions. Likewise, the position of the contraceptive ring may be different in pCT and CBCT due to complex organ motion, which can affect the training process. Thus, we used the threshold method to identify these pixels in the pCT images as much as possible and replaced them with the pixel value located in the same position in the CBCT images. In order to maintain a consistent image size, the dataset acquired from Varian Trilogy were resized to match the size of the dataset acquired from Elekta Axesse using the neighborhood interpolation algorithm. Subsequently, all the images were cropped to a size of 400 × 400 in order to reduce computation resources for unnecessary backgrounds. Finally, the range of HU values was adjusted to [−1000, 1500] and transformed into a linear attenuation coefficient using the following Formula (1):(1)$$\mu_{i}=\frac{{HU}_{i}*\mu_{water}}{1000}+\mu_{water}\;$$
where $\mu_{i}$ is the linear attenuation coefficient of the *i*-th pixel, ${HU}_{i}$ is the CT number of the *i*-th pixel, and $\mu_{water}$ is the linear attenuation coefficient of water. The flow diagram of our image pre-processing is shown in Figure 2.

### 2.2. Network Architecture

The deep learning network was utilized to synthesize 2D pseudo-CT images from input CBCT images. The model established in this study was based on an encoder–decoder architecture. The overall structure of our model is illustrated in Figure 3. U-Net [53] has been successfully applied to medical image synthesis tasks and has achieved promising results [54,55,56,57]. Inspired by this, we adopted U-Net as our baseline model and integrated residual blocks [58] into the model. The model consists of one conventional convolution block, four residual blocks, four up-sampling residual blocks, and one output layer.

The convolution block consists of two convolutional layers with a kernel size of 3 × 3, a stride of 1, and a padding of 1. The residual block differs from the convolution block in that it has an additional branch, which is constructed as identity mapping to simplify the learning process and alleviate the problem of gradient vanishing. Residual learning has already been proven to be effective and has been applied to many tasks related to medical images [44,45,59].

Each of the four up-sampling residual blocks consisted of a deconvolutional layer and a residual block. The deconvolutional layer with a kernel size of 2 × 2 and a stride of 2 was used to progressively recover the output size of the feature maps. Then, the up-sampled feature maps were concatenated with feature maps from the encoder via skip connections to accelerate the training process and preserve the structural details of the input images. In order to accelerate and stabilize the training process, batch normalization [60] was adopted to standardize the inputs after each convolution operation, except for the last layer. Finally, a convolutional layer with a kernel size of 1 × 1 and a stride of 1 was employed to output the synthetic CT images based on the extracted features.

### 2.3. Hierarchical Training Strategy

In this study, we adopted the hierarchical training strategy [61] during the training process. The hierarchical training strategy has been successfully applied to semantic segmentation tasks [62] and classifications [61]. Firstly, we trained the residual U-Net with a coarse resolution, aiming to train the model to learn fundamental image content representations and ordinary image features. Then, we used the previous stage as a starting point to continuously train the model with a finer resolution in subsequent stages to help the model learn the presentation of more complex image features and capture finer details. Here, our model was trained in a progressive manner, with the training process divided into three stages.

In order to generate a multi-resolution image dataset for hierarchical training, we down-sampled the original CBCT images and pCT images by a factor of two. After down-sampling, the number of image pixels was halved. To match the input size of our model, the down-sampled images were then up-sampled to recover to the initial image size. Lastly, we obtained three image datasets with different resolutions. The flow diagram and image demos are shown in Figure 4.

### 2.4. Loss Function

In order to synthesize CT images with both accurate HU values and clear tissue edges, the loss function of our model included three parts: weighted MAE, MS-SSIM loss, and gradient difference loss. MAE is frequently used as a loss function to constrain the overall absolute difference between output and ground truth. However, training the model with the conventional MAE loss function may result in a focus on reducing the difference in regions with high HU values (e.g., bone) while ignoring regions with low HU values (e.g., soft tissues). To address this potential issue, we modified the MAE loss function into a weighted MAE by separately calculating the difference for regions of high HU values and remnant regions. Concretely, the weighted MAE is defined as follows:(2)$$Weighted\;MAE=\frac{1}{N}\sum_{i=0}^{N}\left|y_{high,\;i}-{\tilde{y}}_{high,\;i}\right|+\omega \frac{1}{M}\sum_{i=0}^{M}\left|y_{other,\;i}-{\tilde{y}}_{other,\;i}\right|\;$$
where *N* and *M* denote the total pixel number of regions of high HU values and other regions; $y_{high,\;i}$ and ${\tilde{y}}_{high,\;i}$ are the *i*-th pixel values of the ground truth and the output image in the region with high HU values; and $y_{other,\;i}$ and ${\tilde{y}}_{other,\;i}$ denote the *i*-th pixel values of the ground truth and the output image in remnant regions. Here, we select 500 HU as the threshold to separate the two regions and set the parameter ω to 5.

The structural similarity index (SSIM) [63] is a widely applied metric for evaluating the perceptual quality of images, and the SSIM loss is defined as follows:(3)$$SSIM\;Loss=1-\frac{\left(2\mu_{x}\mu_{y}+C_{1}\right)\left(2\sigma_{xy}+C_{2}\right)}{\left(\mu_{x}^{2}+\mu_{y}^{2}+C_{1}\right)\left(\sigma_{x}^{2}+\sigma_{y}^{2}+C_{1}\right)}\;$$
where $\mu_{x}$ and $\mu_{y}$ and $\sigma_{x}$ and $\sigma_{y}$ denote the mean value and standard deviation of *x* and *y*, respectively; $\sigma_{xy}$ is the covariance of *x* and *y*; $C_{1}$ and $C_{2}$ are constant to prevent division by zero ($C_{1}={\left(0.01\times data\;range\right)}^{2}\;$ and $C_{2}={\left(0.03\times data\;range\right)}^{2}$). MS-SSIM is an extension of SSIM, incorporating multi-scales for measuring the similarity between images. Here, we use MS-SSIM loss to constrain the optimization process, providing a more comprehensive assessment of the agreement between predicted output and ground truth.

Gradient difference loss is used as an extra term to keep strong gradients and enhance edge sharpness during optimization. It is defined as follows:(4)$$Gradient\;Difference\;Loss={\left|\left|{\nabla Y}_{x}-{\nabla \tilde{Y}}_{x}\right|\right|}^{2}+{\left|\left|{\nabla Y}_{y}-{\nabla \tilde{Y}}_{y}\right|\right|}^{2}$$
where Y and $\tilde{Y}$ denote the ground-truth image and synthetic pseudo-CT image, respectively.

In summary, the complete loss function of our model is as follows:(5)$$\begin{array}{ll} Loss= & \lambda_{1}\cdot Weighted\;MAE+\lambda_{2}\cdot MS-SSIM\;Loss\\ & +\lambda_{3}\cdot Gradient\;Difference\;Loss\; \end{array}$$
where $\lambda_{1}$, $\lambda_{2}$, and $\lambda_{3}$ were selected as 1, 3, and 2, respectively.

### 2.5. Evaluation Metrics

In this study, we evaluated the performance of the trained model with three metrics, namely, MAE, SSIM, and peak signal to noise ratio (PSNR). MAE is used to linearly compare the pixel-wise mean absolute difference between synthetic pseudo-CT and ground truth across the whole image. MAE is defined as follows:(6)$$MAE=\frac{1}{N}\sum_{i=1}^{N}\left|Y_{i}-{\tilde{Y}}_{i}\right|$$
where $Y_{i}$ and ${\tilde{Y}}_{i}$ denote the *i*-th pixel value of ground truth and synthetic pseudo-CT. SSIM is used to measure the image similarity between synthetic pseudo-CT and ground truth based on luminance, contrast, and structural information [63]. SSIM is defined as the second term of Formula (3), and here we will not repeat the details. The last quantitative metric is PSNR and it is defined as follows:(7)$$PSNR=20\cdot \log_{10}\left(\frac{{MAX}_{I}}{\sqrt{\frac{1}{XY}\sum_{x=0}^{X-1}\sum_{y=0}^{Y-1}{\left|I_{gt}\left(x,y\right)-I_{syn}\left(x,y\right)\right|}^{2}}}\right)$$
where $I_{gt}(x,y)$ and $I_{syn}(x,y)$ represent the gray value at coordinates (x,y) for the ground-truth and synthetic pseudo-CT images, respectively; X and Y denote image sizes; ${MAX}_{I}$ represents the maximum gray value in both the ground-truth and synthetic pseudo-CT images.

### 2.6. Experiments

In this study, we divided the paired Varian dataset and Elekta dataset into three subsets: the training dataset, the validation dataset, and the test dataset, respectively. The splitting was performed randomly, at a ratio of 6:2:2. The training dataset included 3977 pairs of images from 136 different cervical cancer patients, the validation dataset included 1302 pairs of images from 46 patients, and the test dataset included 1275 pairs of images from 46 patients. An Adam optimizer with a learning rate of 0.001 was utilized and a batch size of 8 was set in this study. All network training, validation, and testing were performed using the Nvidia GeForce RTX 3090 GPU equipped with 24G memory. The code was implemented utilizing the PyTorch Library in Python.

The effectiveness of our network was demonstrated from three perspectives. First, we compared the performance of our model with the RCNN architecture previously used in [45]. They adopted the encoder–decoder architecture as well. Then, we conducted ablation experiments to validate the efficiency of the combined loss function and the hierarchical training strategy: (1) by optimizing the network with different loss functions (weighted MAE only, weighted MAE with gradient difference loss, and weighted MAE with MS-SSIM loss) and comparing their performance; (2) by comparing the performance with models trained without the hierarchical training strategy. Finally, we compared the performance with models trained using a single dataset instead of a hybrid dataset.

## 3. Results

### 3.1. Qualitative Results

In this study, we trained our model in three stages and the total number of epochs was chosen to be 100, including 50 epochs for stage 1, 30 epochs for stage 2, and 20 epochs for stage 3. After optimizing the model weights, we tested its performance on all the test cases. Figure 5 and Figure 6 provide a qualitative comparison of the results obtained from our proposed model and the RCNN model in three randomly selected examples from the test dataset. The top row of Figure 5A–C displays the pCT, original CBCT, and synthetic CT generated by our model and the RCNN model, respectively. Comparing the synthetic CT generated by our model with pCT and original CBCT, the artifacts present in the original CBCT are clearly suppressed and the overall image quality is significantly enhanced. The image quality of the synthetic CT generated by the RCNN model also exhibits improvement, but it appears blurred and fails to maintain uniformity in the same tissues. In order to demonstrate the enhancement in different models, the difference maps were calculated by subtracting the synthetic CT from the pCT. These difference maps are shown in the bottom row of Figure 5A–C, respectively. Darker colors indicate greater differences. It can be observed that the HU value differences between the synthetic CT generated by our model and the pCT are minimal in regions such as the bladder and muscle, in comparison with those generated by the RCNN model and the original images.

Figure 6 demonstrates the HU value histogram of the synthetic CT and the line profiles of HU values passing through the dotted line. The line profiles of the synthetic CT generated by our model indicate that the HU values in the bladder region and bone region have been corrected to be closer to the HU values in the pCT. In comparison to the RCNN model, our model could improve the HU value accuracy and generate images with more similar trends in terms of HU value distribution corresponding to pCT.

### 3.2. Quantitative Results

Table 1 and Figure 7 provide a quantitative comparison of the image quality between the CBCT images and synthetic CT images generated by our model and the RCNN model across the entire test dataset. Compared to the original CBCT images, the MAE between the pCT and synthetic CT generated by our model decreased from 50.02 HU to 10.93 HU, achieving an improvement of 64.21% (*p*-value < 0.001). In terms of PSNR, the synthetic CT images achieved an increase of 26.95% from 27.79 dB to 32.74 dB (*p*-value < 0.001). The SSIM also improved by 19.27%, increasing from 0.77 to 0.90 (*p*-value < 0.001). The RCNN model could also improve the image quality to some extent, achieving mean values of 19.28, 0.82, and 32.03 for MAE, SSIM, and PSNR, respectively. Compared with the RCNN model, our model performed better on all evaluation metrics. 

### 3.3. Ablation Experiment Results

Table 2 summarizes the results of the ablation experiments. All the ablation models were built on the same architecture and trained with the same dataset using the hierarchical training strategy. The first model using weighted MAE as the loss function obtained the highest MAE of 13.77 HU among all the models. By introducing gradient difference loss or MS-SSIM loss into the loss function, the models tended to yield smaller MAE values. However, no significant improvement was observed in terms of SSIM, and there was even a slight decrease. The introduction of the gradient difference loss and MS-SSIM loss was intended to sharpen the edges and enhance image clarity. Nonetheless, this performance could potentially be impacted by the noise and artifacts stemming from the original CBCT. This loss function might identify the noise and artifacts as anatomical structures improperly and try to preserve them during the optimization process, resulting in no significant improvement in terms of PSNR and SSIM. Overall, the combination of three losses resulted in the best performance.

Next, we compared the performance of models trained with different strategies, namely, single-stage training and hierarchical training. Table 3 presents the comparison between different training strategies in all evaluation metrics. The model trained using the hierarchical training strategy achieved a reduction in MAE from 13.57 HU to 10.93 HU (*p*-value < 0.001) compared to the model trained using the single-stage training. The improvement was also evident in terms of PSNR values (*p*-value < 0.001).

### 3.4. Comparison among Different Datasets

In order to investigate the effect of different datasets, we trained two additional models using only the Varian dataset or the Elekta dataset. Table 4 summarizes the evaluation results between the models trained with different datasets. The model trained with the Varian dataset performed better specifically on the same Varian test dataset, with an MAE of 8.77 HU, a PSNR of 38.20 dB, and an SSIM of 0.93, whereas the model trained with the hybrid dataset achieved lower MAE and higher SSIM and PSNR on the same Elekta test dataset compared with specific model trained with the Elekta dataset. This could be attributed to the small size of the Elekta dataset. The specific model for the Elekta dataset was trained using only 1284 paired images, which makes proper generalization more challenging.

## 4. Discussion

In this study, we developed a deep learning-based framework to alleviate the artifacts and enhance the quality of CBCT images for further clinical applications. The experimental results illustrated that the model could efficiently generate synthetic pseudo-CT images with fewer artifacts and a more accurate distribution of HU values compared with the original CBCT images. As mentioned in the introduction, Cycle-GAN [39,43,44] has been frequently used in image synthesis tasks because it can be trained with unpaired image data. However, it is challenging to obtain a balance between the generator and the discriminator, and as a result the model is difficult to train. In addition, it requires more computational resources and needs a long training time. Hence, our model was based on U-Net instead of Cycle-GAN. U-Net has also been widely applied in medical image processing tasks [64,65,66]. It is based on the encoder–decoder architecture. The skip connection can mitigate the information loss during the encoding process and promote information propagation in the network [53]. We incorporated residual learning into the model to enhance the model’s expressiveness and alleviate the vanishing gradient problem.

A hybrid loss function, consisting of weighted MAE, gradient difference loss, and MS-SSIM loss, was utilized to constrain the optimization process. Different from conventional MAE, a weighted MAE was designed to prevent the model from focusing excessively on learning the mapping between high-pixel-value regions. The effectiveness of weighted MAE has been proven by Yang et al. [38]. We also attempted to relieve the image blurriness issue by utilizing MS-SSIM loss and sharpen the edges of soft tissues by introducing gradient difference loss. According to the results of the ablation experiments, we could conclude that it is effective to train the model with the hybrid loss function, which reduced the MAE from 13.77 HU to 10.93 HU, and also led to a slight improvement in PSNR.

The hierarchical training strategy was adopted to train our model. Instead of training the model only in a single stage, we trained our model from a coarse resolution to a fine resolution stage by stage. In the first stage, low-resolution images were used to train the model to learn fundamental image representations. Then, we continually used images with finer resolutions to fine-tune the previous model to help the model capture more details. In order to demonstrate the effectiveness of this training strategy, we also conducted comparative experiments. We trained another model with the same architecture, but only trained with high-resolution images. As shown in Table 3, the model trained using the hierarchical training strategy outperformed the single-stage-trained model in all the evaluation metrics.

Numerous deep learning-based frameworks have been proposed to enhance the quality of CBCT images [38,39,42,43,67]. Wu et al. [45] used an RCNN to improve the accuracy of HU values of CBCT for prostate cancer patients. They obtained the best improvement in terms of MAE, with a decrease of 85.20%. Given that their model was also based on the encoder–decoder architecture, we attempted to compare the performance of our model with theirs. Since their code was private, we reproduced their model on the basis of the description in their papers and trained it with our dataset. The comparison results are presented in Table 1, Figure 5, and Figure 6. Our model outperformed theirs both qualitatively and quantitatively. Furthermore, clear tissue edges could be observed in the images generated by our model. Their model was proposed for prostate cancer patients and it may not work well for our dataset given the more complex anatomy of cervical cancer patients. Rossi et al. [67] also employed U-Net to enhance CBCT image quality, achieving a decrease in MAE from 93.30 HU to 35.14 HU, an increase in PSNR from 26.70 dB to 30.89 dB, and an increase in SSIM from 0.89 to 0.91. Compared to their results, our model achieved better improvements in all metrics. Sun et al. [44] proposed a 3D Cycle-GAN to synthesize pseudo-CT images for cervical cancer patients. They reported an MAE of 51.62 HU, an SSIM of 0.86, and an increase in PSNR from 27.15 dB to 30.70 dB. As for our study, the PSNR increased from 27.79 dB to 33.91 dB. In terms of MAE, it decreased from 50.02 HU to 10.93 HU. We also achieved a higher value for SSIM with an average value of 0.90. Zhao et al. [67] reported the largest improvement in terms of SSIM, increasing from 0.44 to 0.81. However, they used MV CBCT instead of kV CBCT as the input of Cycle-GAN. It was not appropriate to directly compare our results with theirs given that the image qualities of MV CBCT and kV CBCT were essentially different.

The quality of the dataset can significantly impact the performance of the model. In our study, we trained the model with the hybrid dataset in order to improve the generalizability and robustness of the model. The hybrid dataset consisted of images acquired on different linear accelerators from different vendors. In order to investigate the effect of different datasets, we trained specific models for the Varian dataset and the Elekta dataset, respectively. According to the evaluation metrics, our model exhibited inferior performance compared with the specific model for the Varian dataset when using the same test dataset. This may be attributed to different distributions of artifacts and noise in the Varian dataset and the Elekta dataset. When using the hybrid dataset, the model intended to maintain a balance, so the performance was inferior compared with the specific model. Overall, the difference was not particularly obvious in terms of MAE and SSIM, demonstrating the robustness and generalizability of our model to some extent. But for the specific model trained with the Elekta dataset only, the performance became worse. This might be due to the small size of the training dataset, which only includes 29 patients. Better performance may be achieved when training the specific model with a larger Elekta dataset.

According to our promising results, the developed model has the potential to synthesize pseudo-CT images from CBCT images for adaptive radiotherapy applications. However, there are still some limitations in our study. First, the model was developed on the basis of 2D images of a particular pelvic region, limiting the model to leverage spatial information. Moreover, the synthetic CT images generated by the 2D model may fail to maintain anatomical continuity because each slice was independently generated. In the future, we will explore the effectiveness of this framework for 3D images. Meanwhile, we will integrate our framework into Cycle-GAN to develop an unsupervised model, reducing the requirement for paired images. Second, we only evaluated the performance of our model in terms of image similarity. Next, we will comprehensively evaluate the performance of the pseudo-CT images generated by our model in downstream tasks such as auto-segmentation and dose calculation [26,40,68]. Lastly, this study only used the hybrid dataset collected from two linear accelerators, and we will further investigate its performance on multi-center datasets.

## 5. Conclusions

In this study, we successfully developed a deep learning-based framework to synthesize pseudo-CT images from CBCT images for cervical cancer. The model has been shown to generate synthetic CT images with enhanced quality and more accurate HU values as compared to previous models. Particularly, the edges of tissues were well preserved with the constraint of the combined loss function. To our best knowledge, this is the first model built with a multi-vendor dataset and it achieved promising results. It can be further explored in multi-center datasets to promote its application in clinics.

## Figures and Tables

**Figure 1 cancers-15-05479-f001:**
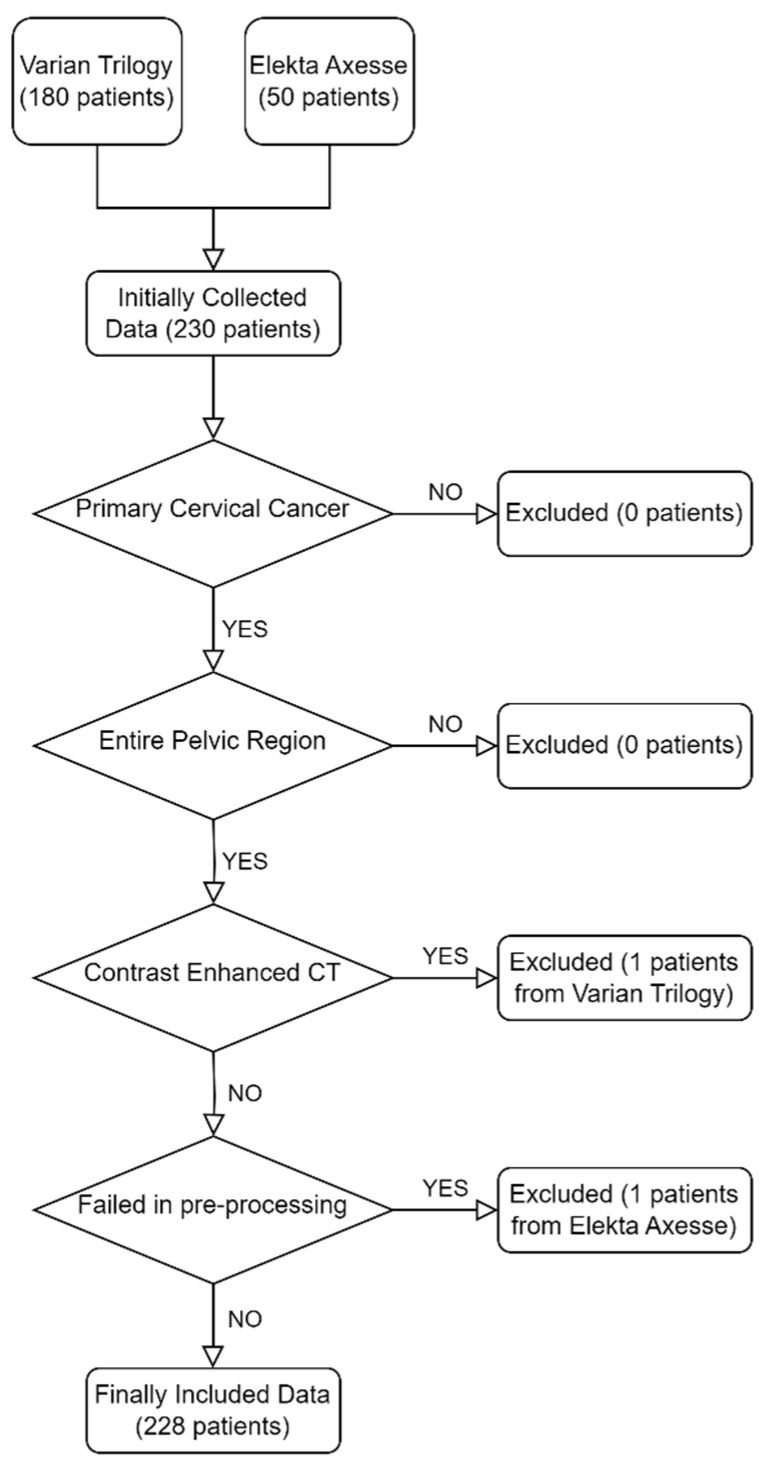
Criteria of data selection. Initially, 180 and 50 patients were collected from Varian Trilogy and Elekta Axesse, respectively. After evaluation, 228 patients were enrolled in this study. One patient was excluded due to contrast agent usage when scanning the planning CT and another one was excluded due to failing in image pre-processing.

**Figure 2 cancers-15-05479-f002:**
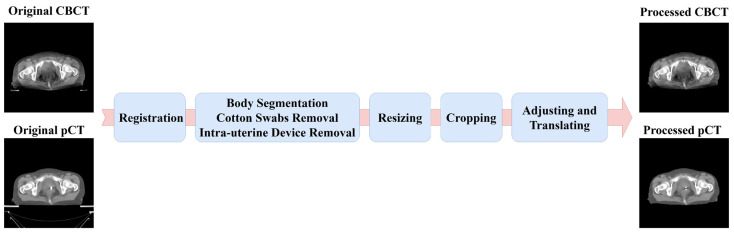
Flow chart of image pre-processing. The left are the original images and on the right are the processed images used to train the model. The window level/width was set to [−100, 400] HU. The whole image pre-processing included registration, removal of non-anatomical structures, resizing, cropping, adjusting, and translating.

**Figure 3 cancers-15-05479-f003:**
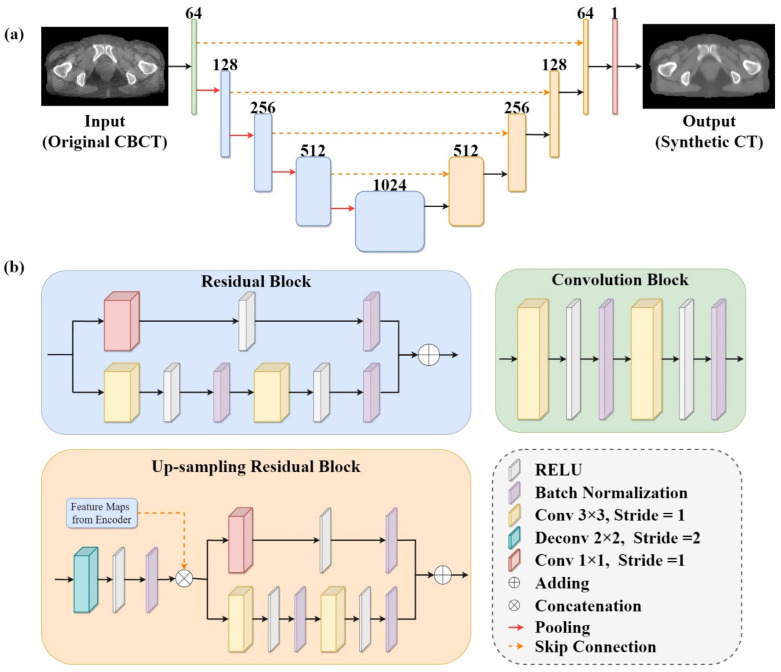
Framework of deep learning model used in this study: (**a**) The overall architecture of the model. The model was developed on the basis of U-Net with residual blocks. The green box represents the convolution block, the blue box represents the residual block, the light orange box represents the up-sampling residual block, and the red box represents the convolution layer with filter size of 1 × 1. The number at the top of box is the filters in the layer. (**b**) Details of the convolution block, the residual block, and the up-sampling residual block shown in (**a**).

**Figure 4 cancers-15-05479-f004:**
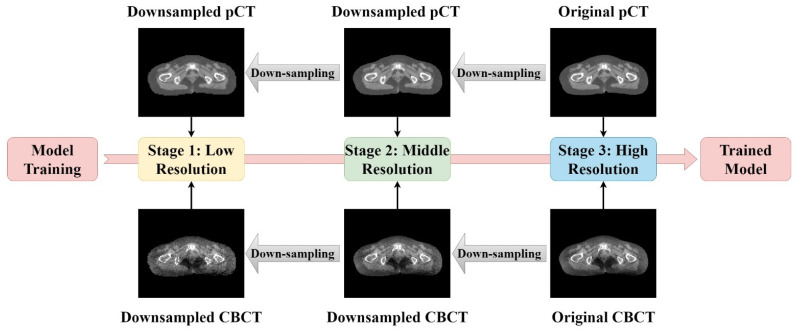
Illustration of the hierarchical training strategy. A total of three stages were included in this study. Except for the first stage, the other two stages were optimized on the basis of the previous stage. The multi-resolution image dataset for hierarchical training was generated by down-sampling the original CBCT images and pCT images by a factor of two.

**Figure 5 cancers-15-05479-f005:**
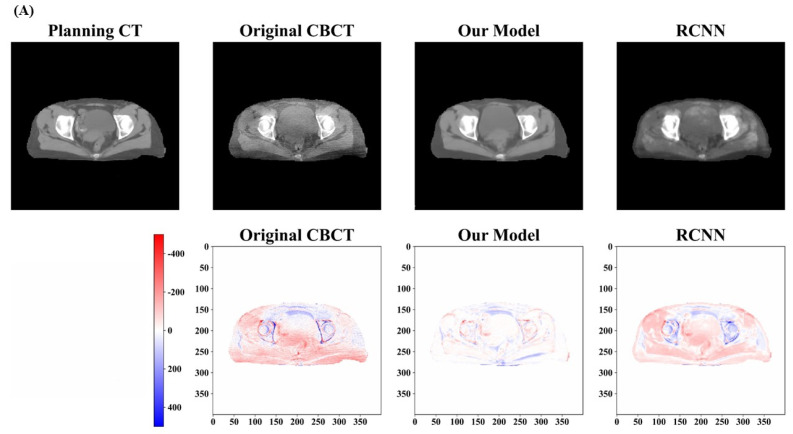
Qualitative results of pseudo−CT images generated by different methods. Three randomly selected test cases are demonstrated in (**A**–**C**). The upper row shows the pCT, CBCT, and synthetic CT generated by our model and the RCNN model from the left to right. The bottom shows the difference maps for synthetic CT images when compared with pCT. The difference of HU values between synthetic CT generated by our model and pCT is minimal and the artifacts are obviously suppressed in all the three test cases.

**Figure 6 cancers-15-05479-f006:**
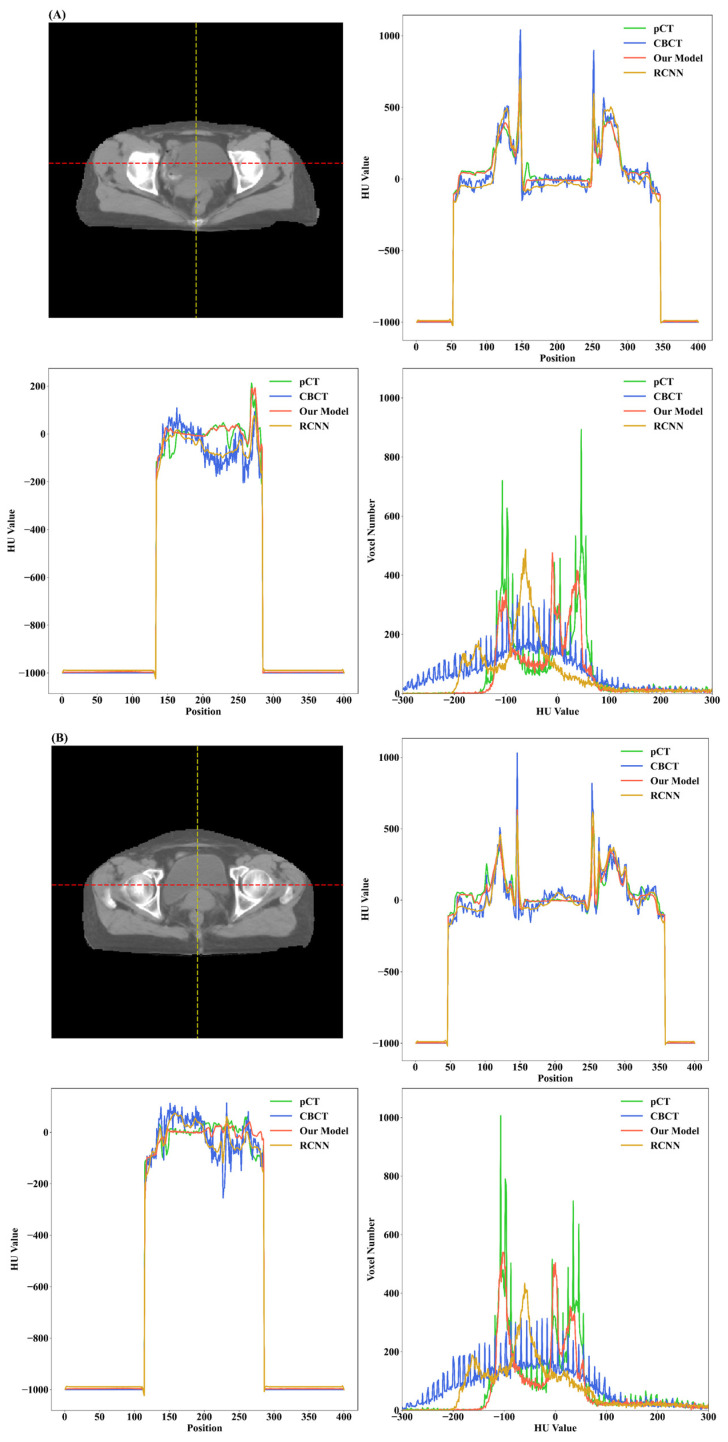
HU line profiles and HU value histogram of test cases shown in Figure 5. For (**A**–**C**), the upper right is the HU line profile passing through the red dotted line on the left image for pCT, CBCT, and synthetic CT produced by our model and the RCNN model, respectively. The bottom left is the line profile passing through the yellow dotted line on the left image. The bottom right is the HU value histogram for the test case. The images generated by our model exhibit more similar trends in terms of HU value distribution corresponding to pCT when compared with the RCNN model and CBCT images.

**Figure 7 cancers-15-05479-f007:**
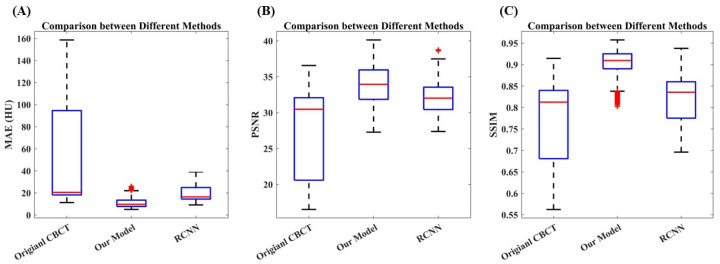
Statistical analysis of quantitative results: (**A**) The comparison between different methods in terms of MAE. (**B**) Comparison of PSNR. (**C**) Comparison of SSIM. The synthetic CT images achieved improvement in all evaluation metrics compared with the original CBCT images. In addition, the established model outperformed the RCNN model.

**Table 1 cancers-15-05479-t001:** Comparison of MAE, SSIM, and PSNR of the synthetic images generated by different models and CBCT images, utilizing the planning CT images as the ground truth.

	CBCT	Our Model		RCNN	
	Value	Value	Improvement	Value	Improvement
MAE (HU)	50.02	10.93	64.21%	19.28	61.45%
SSIM	0.77	0.90	19.27%	0.82	7.029%
PSNR (dB)	27.79	33.91	26.95%	32.03	15.25%

Abbreviations: CBCT = cone beam computed tomography; RCNN = deep convolution neural networks with residual blocks; MAE = mean absolute error; SSIM = structural similarity index; PSNR = peak signal to noise ratio.

**Table 2 cancers-15-05479-t002:** Quantitative results of synthetic CT images produced by models trained with diverse loss functions.

	MAE (HU)	SSIM	PSNR (dB)
Weighted MAE only	13.77 ± 15.60	0.90 ± 0.0011	33.72 ± 6.71
Weighted MAE with GDL	12.92 ± 18.76	0.89 ± 0.0013	33.64 ± 6.92
Weighted MAE with MS-SSIM Loss	11.38 ± 16.67	0.90 ± 0.0010	33.89 ± 6.37
Our Model	10.93 ± 16.76	0.90 ± 0.0010	33.91 ± 6.88

Abbreviations: MAE = mean absolute error; SSIM = structural similarity index; PSNR = peak signal to noise ratio; GDL = gradient difference loss; MS-SSIM = multi-scales structural similarity index.

**Table 3 cancers-15-05479-t003:** Quantitative results of synthetic CT images produced by models trained with different training strategies.

	MAE (HU)	SSIM	PSNR (dB)
Single-Stage Training	13.57 ± 14.44	0.90 ± 0.0008	32.74 ± 3.36
Hierarchical Training	10.93 ± 16.76	0.90 ± 0.0010	33.91 ± 6.88

Abbreviations: MAE = mean absolute error; SSIM = structural similarity index; PSNR = peak signal to noise ratio.

**Table 4 cancers-15-05479-t004:** Comparison of synthetic CT images produced by model trained with different datasets.

Training Dataset	Test Dataset	MAE (HU)		SSIM		PSNR (dB)	
		CBCT	sCT	CBCT	sCT	CBCT	sCT
Varian Dataset Only	Varian Dataset	19.71	8.77	0.83	0.93	31.58	38.20
Elekta Dataset Only	Elekta Dataset	120.41	21.08	0.63	0.81	19.23	30.40
Hybrid Dataset	Varian Dataset	19.71	8.81	0.83	0.92	31.58	35.08
	Elekta Dataset	120.41	15.86	0.63	0.87	19.23	31.19

Abbreviations: MAE = mean absolute error; CBCT = cone beam computed tomography; sCT = synthetic CT; SSIM = structural similarity index; PSNR = peak signal to noise ratio.

## Data Availability

The data presented in this study are available in this article. Further inquiries can be directed to the corresponding author.

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
