# Peer review of "CBCT-to-CT Synthesis for Cervical Cancer Adaptive Radiotherapy via U-Net-Based Model Hierarchically Trained with Hybrid Dataset"

_cancers, 2023, doi:10.3390/cancers15225479_

Round 1
Reviewer 1 Report
Comments and Suggestions for Authors
1- The resolution of figures (such as Figure 6) is not clear and should be improved.
2- For equation number 1 please deleted one addition number.
3- Provide some demographic information of patients in the text.
4- The manuscript has no appropriate description of the results.
5- Figure legends should be provided self-explanatory in detail. Also, all abbreviations used in tables should be defined.
6- Moderate editing of English language grammar and spelling is required.
Comments on the Quality of English Language
6- Moderate editing of English language grammar and spelling is required.
Author Response
We thank you for your time and detailed evaluation of this manuscript. We have read your comments carefully and made the corrections accordingly in the revision. The following is the point-by-point response. The original comments are in black, our responses are in blue.
1- The resolution of figures (such as Figure 6) is not clear and should be improved.
Thanks for the comments and careful review of this manuscript. We have improved the resolution of figures.
2- For equation number 1 please deleted one addition number.
Thanks for your comments. We have deleted one addition number.
3- Provide some demographic information of patients in the text.
Thanks for your comments. We have added some demographic information of patients in the part of Data Collection.
4- The manuscript has no appropriate description of the results.
Thanks for your comments. We have revised the Results section to better describe the results and improved the readability of the results.
5- Figure legends should be provided self-explanatory in detail. Also, all abbreviations used in tables should be defined.
Thanks for your comments. We have revised the figure legends to better demonstrate the figures and also added annotation for abbreviations used in each table.
6- Moderate editing of English language grammar and spelling is required.
Thanks for your comments. We have read through the whole text and improved the language, grammar, and spelling.
Reviewer 2 Report
Comments and Suggestions for Authors
The authors have presented the work titled as "CBCT-to-CT synthesis for cervical cancer adaptive radiotherapy via U-Net based model hierarchically trained with hybrid dataset". The authors must carefully look for minor mistakes and typos mistakes and implement the below mentioned comments.
Figure 1 needs be of better quality and more detailed annotation in the figure and figure legend.
section 2.1.2. image preprocessing, I think authors needs to look properly and how the segmentation step could be bypassed??? Figure 2 is representative but author should look over my question on segmentation.
I appreciate figure which represent the main goal of manuscript.
I am completely surprised that there is no references in discussion and in introduction also less relevant references are present. Please use more and relevant citations. write discussion in professional way otherwise I cannot be that much positive.
Please look carefully through out the manuscript about the typos mistakes like in last reference what is "55."?
Comments on the Quality of English Language
Minor english issues should be addressed.
Author Response
We thank you for your time and detailed evaluation of this manuscript. We have read your comments carefully and made the corrections accordingly in the revision. The following is the point-by-point response. The original comments are in black, our responses are in blue. The corresponding modifications to the manuscript were highlighted.
The authors have presented the work titled as "CBCT-to-CT synthesis for cervical cancer adaptive radiotherapy via U-Net based model hierarchically trained with hybrid dataset". The authors must carefully look for minor mistakes and typos mistakes and implement the below mentioned comments.
Thanks for the comments and careful review of this manuscript.
Figure 1 needs be of better quality and more detailed annotation in the figure and figure legend.
Thanks for the comments. We have improved the image figure quality, and added more detailed annotation in the figures and figure legends.
section 2.1.2. image preprocessing, I think authors needs to look properly and how the segmentation step could be bypassed??? Figure 2 is representative but author should look over my question on segmentation.
Thanks for the comments. We have revised the content of Figure 2 to better demonstrate the image pre-processing. In our study, we only need to remove the non-anatomical structures including the coach, the cotton swabs and intra-uterine device. For other anatomical structures are all preserved in the images. During the process of removing coach, we segmented the body with the tool TotalSegmentator developed by Wasserthal et al. As for the cotton swabs and intra-uterine device, we used threshold-based methods to remove them. We have revised accordingly.
I appreciate figure which represent the main goal of manuscript.
I am completely surprised that there is no references in discussion and in introduction also less relevant references are present. Please use more and relevant citations. write discussion in professional way otherwise I cannot be that much positive.
Thanks for the comments. We have revised the Introduction section and the Discussion section and cited more relevant references.
Please look carefully throughout the manuscript about the typos mistakes like in last reference what is "55."?
Thanks for the comments. We have read throughout the manuscript and corrected the spelling mistakes or typos.